

# A theory on individual characteristics of successful coding challenge solvers

Marvin Wyrich, Daniel Graziotin and Stefan Wagner

Institute of Software Technology, University of Stuttgart, Stuttgart, Germany

## ABSTRACT

**Background**. Assessing a software engineer's ability to solve algorithmic programming tasks has been an essential part of technical interviews at some of the most successful technology companies for several years now. We do not know to what extent individual characteristics, such as personality or programming experience, predict the performance in such tasks. Decision makers' unawareness of possible predictor variables has the potential to bias hiring decisions which can result in expensive false negatives as well as in the unintended exclusion of software engineers with actually desirable characteristics.

**Methods**. We conducted an exploratory quantitative study with 32 software engineering students to develop an empirical theory on which individual characteristics predict the performance in solving coding challenges. We developed our theory based on an established taxonomy framework by *Gregor (2006)*.

**Results**. Our findings show that the better coding challenge solvers also have better exam grades and more programming experience. Furthermore, conscientious as well as sad software engineers performed worse in our study. We make the theory available in this paper for empirical testing.

**Discussion**. The theory raises awareness to the influence of individual characteristics on the outcome of technical interviews. Should the theory find empirical support in future studies, hiring costs could be reduced by selecting appropriate criteria for preselecting candidates for on-site interviews and potential bias in hiring decisions could be reduced by taking suitable measures.

Corresponding author
Marvin Wyrich,
marvin.wyrich@iste.uni-stuttgart.de

## INTRODUCTION

Some well-known technology companies, including Amazon, Facebook, Google and Microsoft, made applicants perform algorithmic programming tasks as part of their technical interview process (*McDowell, 2015*). The performance in these tasks might reveal how good a software engineer is at finding efficient and scalable algorithms to unknown problems and show his or her ability to debug and test a small piece of source code. In the following we refer to these tasks as coding challenges. Although aspects such as interpersonal skills play an important role in technical interviews (*Ford et al., 2017*), coding challenges form an essential part of the interviews and the subsequent evaluation of the candidates.

To help candidates preparing for competitions, technical interviews and coding challenges in general, several books, online guides, practicing websites and experience reports exist (e.g., *LeetCode, 2018*; *Dumitru, 2017*; *McDowell, 2015*; *Mongan, Kindler & Giguère, 2012*). A plethora of material is available that aims to help the reader solve coding challenges successfully.

We can see, however, that some software engineers perform better than others in solving coding challenges (*Google Inc., 2017*) and that this difference does not necessarily come all from preparation. Practitioners report that they met computer science graduates who could not even solve simple programming tasks like the Fizz-Buzz challenge[1] up to developers who aced almost every challenge in the finals of internationally recognized programming competitions.

As with programming performance (*Scacchi, 1995*), individual characteristics are likely to play a very important role in the performance of solving coding challenges. To the best of our knowledge, there is a knowledge gap in the software engineering literature to explain individual factors related to a successful solver.

Addressing the knowledge gap about individual characteristics of successful coding challenge solvers could be favorable for software companies as well. The companies' unawareness of possible predictor variables, such as a candidate's personality, may lead to biased hiring decisions and candidates with actually desired personality traits having less chances of getting hired. Being aware of possible predictor variables could therefore help the company to better preselect candidates at an early stage of the interview process as well as identify ways to improve the current workforce. Furthermore, failure to understand which characteristics make for a great coding challenge solver might bring failure in a technical interview. Job opportunities might get lost.

A knowledge gap in a research discipline, as in the case of coding challenges, requires the construction of theories. *Kajko-Mattsson (2012)* has argued that software engineering research has been suffering from a syndrome that causes researchers to jump from trend to trend. What happens is that isolated, usually small research problems are solved by one or more papers and then authors jump to a completely different issue. As a result, continues *Kajko-Mattsson (2012)*, also echoed by *Johnson, Ekstedt & Jacobson (2012)*, the research community lacks a deep, yet basic understanding of the software development life-cycle and the theory behind all software engineering activities. We support the recent request for developing theories in software engineering and we agree with several authors that theory is what software engineering is missing the most (*Ralph, Johnson & Jordan, 2013*; *Johnson et al., 2013*; *Johnson, Ekstedt & Jacobson, 2012*; *Johnson & Ekstedt, 2015*; *Wohlin, Šmite & Moe, 2015*).

We conducted a study to explore the research question *what individual characteristics make a good coding challenge solver?* By a quasi-experiment, we empirically developed a theory for predicting the influence of such characteristics on the performance in solving coding challenges. We developed and evaluated the construction of the theory using an established framework for theories in information systems by *Gregor (2006)* which we present in the next section.

[1] The Fizz-Buzz challenge is a trivial programming task that is used in interviews to filter out programmers with insufficient programming skills. The task is to write a program that prints the numbers from 1 to 100. But for multiples of three print "Fizz" instead of the number and for the multiples of five print "Buzz". For numbers which are multiples of both three and five print "FizzBuzz" (*Ghory, 2007*).

### Research objectives and contributions

The objective of our research is to construct a theory on how individual characteristics of a software engineer predict his or her performance in solving coding challenges. We contribute the following main results:

- We found significant negative correlations between the affective state of sadness and the performance in solving coding challenges, as well as between the personality trait of conscientiousness and the performance.
- Significant positive correlations were found between variables related to the academic performance and the coding challenge performance, as well as between programming experience and performance.
- These findings were brought together in a Type III theory for predicting (*Gregor, 2006*). Statistically significant results were achieved to offer a theory grounded in data and we offer the theory for testing by future studies.

In the following section we provide a summary of related work on coding challenges and describe the scientific basis of our theory construction. In the 'Methods' section we describe the research methodology in detail, including the design of our study, its sample, used coding challenges, candidates for predictor variables and the analysis procedure. Findings of our study are presented in the 'Results' section, followed by a discussion of the findings, limitations and implications of our study.

## BACKGROUND

A coding challenge (also called programming challenge) is an algorithm and coding problem used to assess a programmer's problem-solving skills. Coding challenges are used in several areas of applications and websites exist that offer different types of coding challenges for learning, practicing, and competing with other users. In most cases, one has to find the most efficient algorithm or any correct algorithm within a limited amount of time. These are the coding challenges relevant to our work. Other types of coding challenges include those that can be solved by completing code to win a game or by writing code that passes all given test cases.

Programming competitions, for example, use coding challenges as tasks for the participants. Such competitions enjoy wide popularity. In 2017, the ACM International Collegiate Programming Contest (ICPC) recorded 46,381 students from 2,948 universities in 103 countries (*Baylor University, 2017*). In the same year, the winner of the Google Code Jam prevailed against more than 25,000 competitors and won a grand prize of $15,000 (*Google Inc., 2017*). It is worth noting that *Bloomfield & Sotomayor (2016)* found most students were not even motivated by the prizes when participating in the ICPC. They understood that participating in programming contests requires skills which are valuable for job interviews where technical questions are asked.

Research, on the one hand, focuses on the design and scoring of coding challenges in programming competitions. We elaborate on these aspects in the 'Methods' section where we describe the selection of coding challenges selected for our study and how we

scored the solutions. On the other hand, existing research includes the usage of coding challenges for educational purposes such as the automated grading of assignments and teaching algorithms. For example, *Urness (2017)* presented the hypothesis that interview question assignments would be beneficial for students because they require intense practice and are more motivating for the students due to their real-world applicability. In a course on data structures, *Urness (2017)* compared the exam performance of students who were taught with long-term programming assignments with that of students who had to solve short-term interview-question assignments throughout the semester. *Urness (2017)* found that students enrolled in the interview question assignment section had a slightly better average score in the final exam and that the interview question assignments were motivating for students.

Technical interviews in particular have not been investigated thoroughly in scientific literature yet. We identified only two recent relevant papers. *Ford et al. (2017)* investigated whether there are company differences in interview criteria and how interviewers interpret criteria for software engineer job candidates. Their research was motivated by an existing mismatch of candidates' expectations of what interviewers assess and what they actually look for in a candidate, which consequently results in lost job opportunities. Coding challenges from *Cracking the Coding Interview* (*McDowell, 2015*) were used as interview questions in mock technical interviews with university students and interviewers from nine different software companies. To evaluate a candidate's performance, interviewers had to fill out an evaluation form for each candidate which included six criteria and an open response section. After the interviews the authors analyzed the evaluation forms to answer the research questions. First, they found consistent interviewers' expectations for candidates among most companies. Second, interviewers care about technical soundness but place an emphasis on interpersonal skills and effective communication. This finding is consistent with the results of previous studies on the demand for soft skills in software engineering. For example, *Matturro (2013)* found that about 70% of job advertisements had at least one soft skill listed as a requirement. Teamwork and communication skills but also analytical problem-solving skills were some of the most demanded skills. The interviewers in *Ford et al. (2017)*'s study noted that candidates had difficulties in communicating their knowledge. However, there is research on how to bridge the gap between what universities teach and what industry demands in terms of interpersonal and communication skills (e.g., *Teles & De Oliveira, 2003*).

Potentially excessive stress and cognitive load due to technical interviews in which candidates have to solve tasks on a whiteboard reinforce bias in hiring practices. For this reason, *Behroozi et al. (2018)* conducted a study on the differences in stress and cognitive load between solving programming tasks on paper versus on a whiteboard. To assess the difference they used eye measures, measured by a head-mounted eye-tracker and computer vision algorithms. Each of the eleven participants completed tasks under each of the two conditions (paper and whiteboard). The authors then conducted debriefing interviews and analyzed the eye tracking data. Preliminary results suggest that problem-solving on the whiteboard results in higher cognitive load compared to solving programming problems

on paper. In addition, participants rated the whiteboard interview as more stressful. Only in the whiteboard setting nervous tics displayed by participants were observed.

Without closer reference to technical interviews, attempts were made to predict how good a developer will perform in a specific environment, based on characteristics that we examine in part in our study. For example, *Borreguero et al. (2015)* developed an index to find virtuous developers, which is based on the activities and contributions of the respective developers in open-source communities. The authors discuss similar work that aims, for example, at finding experts in such communities. Although this is not about experts for solving coding challenges, at least we wanted to mention these related approaches in our work. Further research is needed to evaluate the approach by *Borreguero et al. (2015)* and to be able to compare our results with theirs on which individual characteristics best predict a developer's performance in the respective environments.

### Theory construction and representation

To give a definition of what information systems researchers mean when they deal with *theory*, *Gregor (2006)* proposed a taxonomy of five theory types in information systems research. Each theory type has its own definition and consists of a set of structural components: means of representation, constructs, relationships, scope, causal explanations, testable propositions, and prescriptive statements. The theory for predicting, called *Type III*, "states what will happen in the future if certain preconditions hold" and "has testable propositions but does not have well-developed justificatory causal explanations" (*Gregor, 2006*: 619–620). We constructed a theory for predicting the influence of individual characteristics on the performance in solving coding challenges. With respect to the taxonomy and the relevant components for Type III theories, we ensured to conduct and describe our study accordingly.

## METHODS

To answer the research question and consequentially develop the above-mentioned theory for predicting, we conducted an exploratory quantitative study in which participants had to solve three coding challenges on a computer and fill out questionnaires about their individual characteristics. Exploratory research intends to gain information for further research through exploring a research question. "Exploratory research cannot provide a conclusive answer to research problems [...], but they can provide significant insights to a given situation" (*Singh, 2007*: 64).

### Research design

To motivate potential participants to take part in the study, they were told that the study would last at most 90 min and that it was about coding challenges, similar to those used by several software and technology companies during their interview process. They were also told that the reason for the study is to find out why some software engineers perform better in coding challenges than others. Per slot, one or two participants then were invited to a quiet room where they were provided with an informed consent form and introduced verbally to the study. We used the same set of instructions to make sure that every

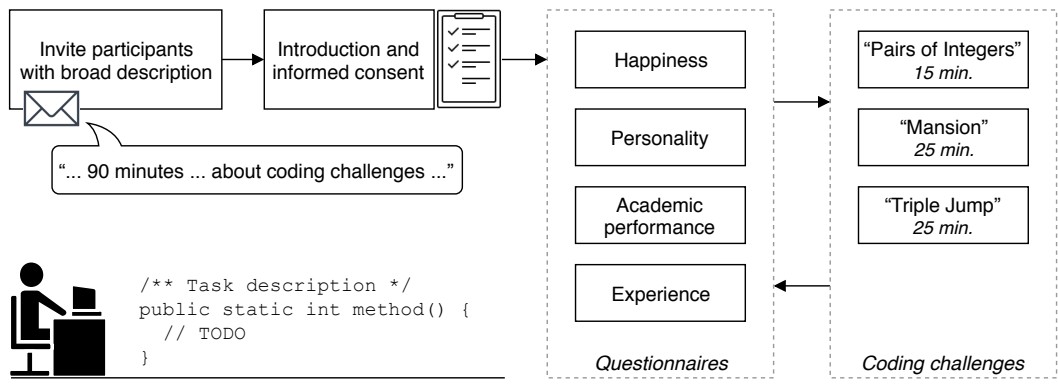

**Figure 1 Schematic representation of the research design.** Four questionnaires on the individual characteristics had to be completed alternating with solving three coding challenges.

participant received the same information. A translated English version of our participation instructions is provided in the paper supplements. After the introduction, participants had the chance to ask questions before they started to fill out the first questionnaire. A schematic representation of the research design is provided in Fig. 1.

Participants had to solve coding challenges implemented with Java on a computer without the use of the Internet or other resources. To make sure that there was no advantage or disadvantage for any participant due to not knowing the used development environment, participants were asked if they were familiar with Eclipse and Java. Each coding challenge had to be solved individually and within a given time. There was a given method signature so that the type of the return value as well as the parameters of this method were defined. It was not allowed to change the method signature in any way. A description of the problem was provided as a comment above the method. We will describe the challenges in greater detail below. The task then was to implement the method with a time-efficient solution to the problem. It was allowed to add private methods if needed and to use methods and data structures of the predefined Java packages. Participants were told that the solutions would be evaluated by correctness and time complexity, which are common judgment criteria for technical interviews (*McDowell, 2015*). While solving a coding challenge, participants were allowed to take notes on paper.

In addition to the given method signature for each challenge, there was also a `main` method with an example call of the method to be implemented. The expected output was provided as well. We provided the example to make it easier for the participants to understand the task and to increase the likelihood that no further questions were necessary while a participant solved a coding challenge. The participants were allowed to run the `main` method and to modify it to add their own test cases if desired. There was no other feedback on the correctness or efficiency of a participant's solution than what the `main` method tested. Participants were told that there was no advantage in submitting a solution before the time was up. If a candidate implemented multiple solutions within the time limit, he or she had to decide which one to submit in the end. When evaluating the solutions

afterwards we only considered the implementation provided in the prescribed method signature. We report our evaluation steps in the 'Analysis procedure' section.

## Participants

All participants of the study were software engineering students of the University of Stuttgart and they had to be at least at the end of the second semester of their bachelor program. The reason for the latter requirement was that at this point a student has the fundamental knowledge of data structures, algorithms, and time complexity that was required for solving the coding challenges in a time-efficient way. In their second semester, software engineering students attend a lecture which is specifically about data structures and algorithms.

The sample consisted of 14 bachelor students at the end of their second semester, seven students who were at least at the end of their fourth bachelor semester and eleven students who studied in the master's program. These students were personally invited by email. In total, 32 participants took part in our study. Five identified as female and 27 as male. The average age of the participants was 21.88 years with a standard deviation of 2.4.

Although limited geographically and culturally, the sample was rich as participants covered the whole spectrum of academic levels. Furthermore, they represent potential participants of technical interviews, that is, fresh BSc and MSc graduates in computer science and software engineering.

## Coding challenges

The criteria for a good task vary depending on the context, target group and what the reasons are for conducting or attempting a coding challenge. For example, an interviewer wants to test a candidate's ability to develop an algorithm, whereas a teacher of a programming course might want to teach time and space complexity. Coding challenges will be selected accordingly.

From what we know from the opinions and experiences of interviewers, we can argue that the existence of the following characteristics of a coding challenge has proven its worth in technical interviews (*McDowell, 2013*; *McDowell, 2017*): A brute-force solution which describes an algorithm that systematically goes through all possible solutions to a given problem should not be the most efficient solution to the problem. The reason is that brute-force algorithms usually are the most obvious way of solving a coding challenge and so they are the first solutions that even a below-average coding challenge solver can come up with. If one reason for developing a coding challenge is to find out if a candidate can think critically about his or her initial solution and how this solution can be optimized, then coding challenges with an inefficient brute-force solution and ways to improve it are most suitable. Again, for interviewers it is important to see the logical thinking process and how the candidate approaches an unknown problem (*McDowell, 2013*). Also, a coding challenge should therefore not just test a single piece of knowledge, for example, a particular programming language feature, except this is what the interviewer aims for. There would be a high chance that some otherwise good coding challenge solvers do not know about this single fact and thus the results become unreliable. More generally, *McDowell (2015)* recommends interviewers to "use hard questions, not hard knowledge" to focus on problem-solving and other skills that cannot be learned quickly at work.

Recommendations from researchers in the field of creating tasks for programming competitions are similar to the characteristics of good coding challenges for technical interviews. For example, *Burton & Hiron (2008)* identified short and easy-to-understand problem descriptions as one criterion for a good task on the International Olympiad in Informatics. Another is the existence of several solutions of varying difficulty and efficiency. Different from technical interviews, programming competition tasks also consider, for example, that tasks should be fun and allow participants to have learning experiences (*Dagiene & Futschek, 2008*).

We followed the recommendations of the interviewers and researchers mentioned above and designed our study in a way that a participant only had to interact with his or her computer and that there should be no need for asking for further clarification. This is different from some technical interviews where the challenge description does not provide enough information to find a satisfactory solution and the candidate is expected to ask further questions before attempting to solve the coding challenge. In our study, we only measure the combination of finding and implementing an algorithm to a given coding challenge, and neither explicitly observe how well participants are at understanding problem statements nor how much a participant cares about requirements engineering. Thus, in addition to the characteristics described above, for our study a good coding challenge was not only easily understandable but also unambiguous. We also chose challenges where finding an efficient solution was challenging but the implementation should have been straightforward, because we could not expect every candidate to be familiar with the particularities of the programming language. To avoid making the participants spend too much time on handling edge cases, some limitations to the input parameters were provided in the task description.

The set of coding challenges we chose covers a range of concepts which are commonly required for solving coding challenges. This includes the use of appropriate data structures, an optimization problem, and recursive thinking. In the following we describe each of the three coding challenges. They were presented to the participants in German, which is their native language, to minimize misunderstandings. The time limit given for each challenge was for understanding the task, finding an algorithm, and implementing the algorithm. We piloted the study with a male student in a higher bachelor semester to make sure that the time limit for each challenge was sufficient for the participant to come up with a solution. The test participant was able to solve the first two challenges correctly with some time pressure for the first challenge and no time pressure for the second one. In addition, after the pilot test we showed the candidate possible solutions for the third task and it took only a short time until he understood the proposed solutions. This strengthened our assumption that the tasks can be solved and that they can be solved within the given time. Additionally, after each challenge we assessed whether the participants felt that they were under time pressure. We report all assessed variables in section 'Conceptual model'.

### Challenge 1—Pairs of Integers (15 min)

The first coding challenge, in Listing 1, was considered to be the easiest one, at least when it comes to finding any correct solution to the problem. A brute-force algorithm where

every possible combination of two numbers is tested in two `for`-loops is straightforward and runs in $\mathcal{O}(n^2)$, where n is the size of the integer array. Another conceivable solution with a time complexity of $\mathcal{O}(n \cdot log(n))$ sorts first and then applies a binary search to find matching pairs. Furthermore, by using an appropriate data structure one can come up with a solution that runs in $\mathcal{O}(n)$. Such a solution is given in Listing 2.

### Challenge 2—Mansion (25 min)

The second challenge was taken from the Australian Informatics Olympiad 2007 and is presented in Listing 3. In the `main` method there was a detailed example given including an ASCII art that illustrated the scenario, similar to the example and illustration given in the task description of the AIO task (http://archive.is/E5qEG). One approach of solving this coding challenge is to use a sliding window of length $w$ that covers $w$ houses and is shifted along all the houses in the array. The sum of people living in these $w$ houses is the amount of people living opposite the mansion. Each time when shifting the window, one can either calculate the sum of all houses covered by the window or simply use the last sum and subtract the people from the one house that is no longer covered by the window and adding the number of people of the one house that is now covered by the window. The first solution runs in $\mathcal{O}(n \cdot w)$ and the second solution runs in $\mathcal{O}(n)$, where n is the number of houses.

This problem involves considerable problem translation, as it is a rather abstract task. We wanted to ensure that different common types of coding challenges were represented in our study and this task is a type of challenge that appears in technical interviews (see, e.g., *McDowell (2015)*). To prevent comprehension problems, the problem descriptions were presented to the participants in their mother tongue. During the experiment, the first author of this submission was present in the room to answer any questions that arose. Of the 32 participants, only one person asked for clarification of the mansion task. We have seen no indication that misunderstandings arose.

### Challenge 3—Triple step (25 min)

The third coding challenge is also the hardest one in our study. It is described in the book *Cracking the Coding Interview* (*McDowell, 2015*) and the ability of recursive thinking is beneficial to find an approach to solving this problem. The challenge itself is provided in Listing 5. A simple recursive solution with a time complexity of roughly $\mathcal{O}(3^n)$ is in Listing 6.

An alternative implementation would be a recursive depth-first search for all possible permutations starting from the bottom of the staircase. In either case the time complexity is not ideal as the same subtrees have to be calculated multiple times. For example, in the recursive solution in Listing 6 for $n = 4$ the algorithm calculates the solution for *countWays*(2) twice. One could store the solutions to such a subproblem in an additional memory structure. This reduces the time complexity to $\mathcal{O}(n)$. As we told the participants that each solution is only evaluated by correctness and time complexity, we ignore the differences in space complexity. However, with the iterative solution in Listing 7 one can avoid the need for additional memory and get to a solution with a time complexity of $\mathcal{O}(n)$ as well.

```java
/**
 * In the given array, find the number of integer pairs with a given
 * difference.
 *
 * An example is given in the main method.
 *
 * The following applies:
 *    numbers contains at least two integers
 *    numbers contains no duplicates
 *    dif >= 1
 */
public static int pairCount(int[] numbers, int dif) {
    // TODO
    return 0;
}

public static void main(String[] args) {
    int[] numbers = {1,5,3,6,8};
    int dif = 2;

    // Expected output: 3
    // The pairs with a difference of two are: {1,3} {5,3} {6,8}.
    System.out.println(pairCount(numbers, dif));
}
```

Listing 1: Coding challenge 1

```java
public static int pairCount(int[] numbers, int dif) {
    HashSet<Integer> set = new HashSet<Integer>();
    int count = 0;
    for (int i: numbers) {
        if (set.contains(i + dif)) {
            count++;
        }
        if (set.contains(i - dif)) {
            count++;
        }
        set.add(i);
    }
    return count;
}
```

Listing 2: A solution to coding challenge 1 in $\mathcal{O}(n)$

```java
/**
 * You want to build a mansion along a road. On the other side of the street
 * there are houses, in which a certain number of people live.
 *
 * Your mansion is as long as w houses together.
 *
 * Place your mansion in such a way that on the other side of the street as
 * many people as possible live opposite your mansion.
 * This greatest possible number should be returned by this method.
 *
 * An example is given in the main method.
 *
 * The following applies:
 *    1 <= w <= houses.length <= 100000
 *    The number of people in each house is >= 0
 */
public static int mansion(int[] houses, int w) {
    // TODO
    return 0;
}
```

Listing 3: Coding challenge 2

```
public static int mansion(int[] houses, int w) {
    int count = 0;
    for (int i=0; i < w; i++) {
        count += houses[i];
    }

    int lastWindow = count;
    for (int i=1; i + w <= houses.length; i++) {
        int currentWindow = lastWindow - houses[i-1] + houses[i-1 + w];
        if (currentWindow > count) {
            count = currentWindow;
        }
        lastWindow = currentWindow;
    }

    return count;
}
```

Listing 4: A solution to coding challenge 2 in $\mathcal{O}(n)$

```
/**
 * A child is climbing up a staircase with n steps, and can hop either
 * 1 step, 2 steps, or 3 steps at a time. Implement a method to count how
 * many possible ways the child can jump up the stairs.
 *
 * An example is given in the main method.
 *
 * The following applies:
 *     0 <= n <= 30
 *     if n=0 return 1
 */
public static int countWays(int n) {
    // TODO
    return 0;
}

public static void main(String[] args) {
    int n=3;

    // Expected output: 4
    // These are the possibilities to climb the three stairs:
    // {1,1,1}, {1,2}, {2,1}, {3}
    System.out.println(countWays(n));
}
```

Listing 5: Coding challenge 3

```
public static int countWays(int n) {
    if (n<0) {
        return 0;
    } else if (n == 0 || n == 1) {
        return 1;
    } else {
        return countWays(n - 1) + countWays(n - 2) + countWays(n - 3);
    }
}
```

Listing 6: A recursive solution to coding challenge 3 in $\mathcal{O}(3^n)$

```
public static int countWays(int n) {
    int result = 1;
    int a=1;
    int b=0;
    int c=0;

    for (int i=0; i < n; i++) {
        result = a + b + c;
        c = b;
```

```
        b = a;
        a = result;
    }
    return result;
}
```

Listing 7: An iterative solution to coding challenge 3 in $\mathcal{O}(n)$

## Conceptual model

Based on existing literature, we created a conceptual model to aid our quantitative exploration. We included four constructs related to individuals that are potentially linked to coding challenge performance, namely happiness, experience, academic performance, and personality. We describe each of them in the following, justify their inclusion in our study with relevant literature and describe how we operationalized and measured. The measure of the coding challenge performance is described in the 'Analysis procedure' section.

We provide our conceptual model, variables, and operationalization in Fig. 2. The following subsections describe the candidate constructs as well as a rationale for their inclusion.

### *Happiness*

Before explaining the inclusion of happiness, we need to define it together with the related concept of affect. Under a hedonistic view,[2] happiness is a sequence of experiential episodes (*Haybron, 2001*) and individuals are happy when they experience "an excess of positive over negative affect" (*Bradburn, 1969*: 9). Affect, in turn, has been defined by *Russell (2003)* as "a neurophysiological state that is consciously accessible as a simple, nonreflective feeling that is an integral blend of hedonic (pleasure–displeasure) and arousal (sleepy–activated) values" (p. 147). Affect, in other words, is the basic building block of emotions and moods.

We consider it a sensible choice to have happiness as a candidate predictor for coding challenge performance: happiness and affect in general have been found to positively impact job performance (e.g., *Oswald, Proto & Sgroi (2015)*) and analytic problem-solving (e.g., *Graziotin, Wang & Abrahamsson (2014)*). We have published extensively on the relationship between happiness and software developers' performance while programming (e.g., *Graziotin et al. (2018)*; *Graziotin, Wang & Abrahamsson (2015)*, the latter being a theory of affect and performance while programming). In our studies, we found support for the happy, therefore productive (high-performing) developer.

In our studies, when we had the need to quantitatively assess the happiness of developers (*Graziotin, Wang & Abrahamsson, 2014*; *Graziotin et al., 2017*), we opted for the Scale of Positive and Negative Experiences (SPANE, *Diener et al. (2010)*). SPANE converges to other similar measurement instruments and has been psychometrically validated in several large-scale studies (*Rahm, Heise & Schuldt, 2017*; *Diener et al., 2010*; *Silva & Caetano, 2013*; *Li, Bai & Wang, 2013*; *Sumi, 2014*; *Jovanović, 2015*; *Corno, Molinari & Baños, 2016*; *Du Plessis & Guse, 2017*) including consistency across full-time workers and students (*Silva & Caetano, 2013*).

SPANE assesses how often a participant has experienced several affective states over the past four weeks. Six positive and six negative states are graded on a 5-point scale of

[2]As opposed to Aristotelian *eudaimonia* which is the realization of conducting a satisfactory life full of quality (*Haybron, 2005*).

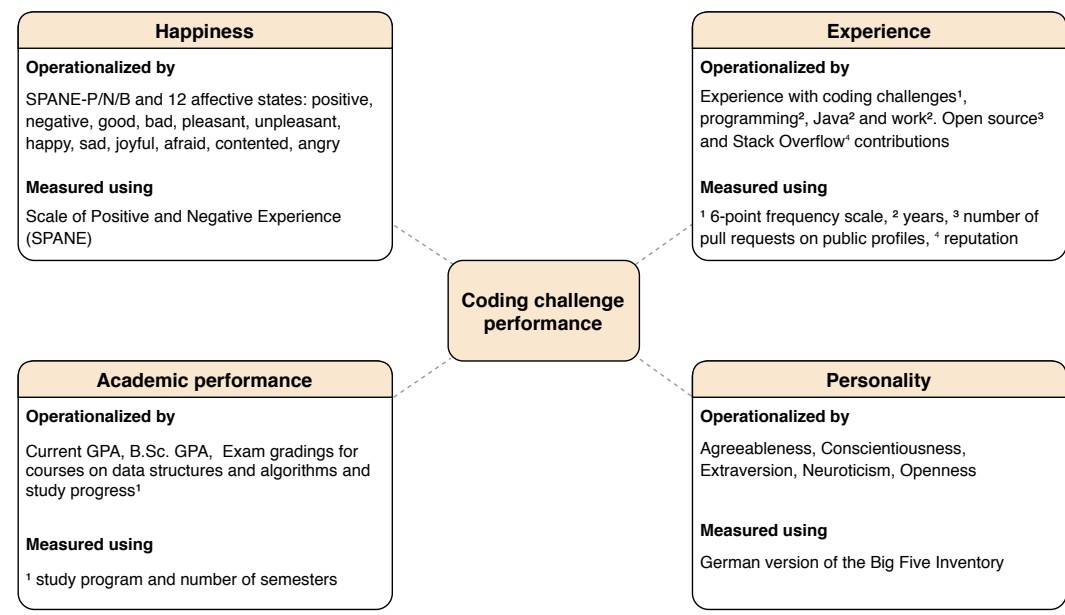

**Figure 2** **Overview of the candidates for predictor variables, their operationalization and measurements.** Investigated relationships with the coding challenge performance are represented as dashed lines. Footnotes for operationalizing variables refer to the used measure for this operationalization within the same rounded rectangle.

frequency. The positive and negative affective scores can be summed to form the SPANE-P(*ositive*) and SPANE-N(*egative*) dimensions, which range from 1 to 30. A subtraction of the SPANE-N from the SPANE-P value results in the *affect balance*, or SPANE-B (*Diener et al., 2010*), dimension which can vary from −24 to 24. A value of 0 indicates a balance of frequency of positive and negative affective experiences, and −24 and +24 indicate the negative and positive extremes, respectively.

### *Experience*

For us, experience is "knowledge, skill, or practice derived from direct observation of or participation in events or in a particular activity" and "the length of such participation"(*Merriam-Webster (2016)*, online). In particular, we were interested in the practical programming-related experience of software engineers and its relation to the coding challenge performance. Programming experience is a multi-faceted construct that is usually expressed in years (e.g., *Mäntylä et al. (2014)*). It is widely accepted that the more experienced software engineers are, the higher their productivity will be (*Siegmund et al., 2014*) and, more interestingly for the present study, their program comprehension abilities (*Siegmund et al., 2014*). This is why we included experience in our set of candidates to predict coding challenge performance.

We operationalized the construct of experience with frequency of coding challenges over the last year, general, Java-related, and professional programming experience in years, but also reputation on StackOverflow.com. We included the latter because developers rely on it, draw on the knowledge from experts at different levels, and developer expertise

is well represented by good questions and answers on it (*Abdalkareem, Shihab & Rilling, 2017*; *Pal, Harper & Konstan, 2012*; *Kumar & Pedanekar, 2016*). For similar reasons, we also operationalized experience with the number of pull requests on GitHub.com as they are positively correlated to several experience measures (*Rahman & Roy, 2014*).

### Academic performance

Academic performance, or academic achievement, is "the extent to which a person has accomplished specific goals that were the focus of activities in instructional environments, specifically in school, college, and university" (*Steinmayr et al. (2014)*, online). While it would seem sensible to expect a positive correlation between academic performance and coding challenge performance, the literature did not provide any strong indication on either side. *Sackman, Erikson & Grant (1968)* neither found a correlation between the performance of experienced programmers and trainee class grades nor between the performance and the score in programming ability tests; however, *Darcy & Ma (2005)* found that participants with a higher academic performance performed better in their programming task. Building upon a non-established truth on academic performance, we included academic performance as a candidate for predictors to coding challenge performance, and we operationalized the construct with the students' GPA score and exam grades for two courses on data structures, algorithms and complexity. The first course is called *data structures and algorithms* and students usually take it in their second bachelor semester. The second course is called *algorithms and complexity* and students usually take it at a later stage of their bachelor studies. The latter course covers algorithm complexity in more detail while the first course gives a practical introduction to the usage of data structures and algorithms for common problems. The two courses have different examiners. We also asked for the current semester in which the students are enrolled at as a way to assess the seniority of the participants.

### Personality

Software engineering research on personality and individual performance can now be considered to be established, mature, yet still relevant and increasing (*Cruz, Da Silva & Capretz, 2015*). No prior research investigated coding challenges and personalities. However, an older study by *Evans & Simkin (1989)* found that a personality trait was a statistically significant explanatory factor for mastering computer concepts. Once again, we had to inspect related work on more generic performance.

In their systematic mapping study on personality research in software engineering, *Cruz, Da Silva & Capretz (2015)* found conflicting evidence on the influence of personality and various conceptualizations of performance of developers.[3] There have been reports of personality being a successful indicator of academic performance (*Layman, 2007*) as well as not being a significant factor predicting academic performance (*Golding, Facey-Shaw & Tennant, 2006*).

Some research conflict was found about individual performance of programmers. Positive relationships between personality traits and programming task performance

[3]We report some examples here but direct the reader to the work by *Cruz, Da Silva & Capretz (2015)* for many more references and interesting points.

(*Shoaib, Nadeem & Akbar, 2009*; *Karimi et al., 2016*) have been found together with no significant relationship between personality and programming performance (*Bell et al., 2009*).

We see a mild indication that personality has an influence on individual performance while developing software and, in the absence of a strong trend, we include the construct in our conceptual model in the hope to shed some more light on the matter.

Even though the Myers–Briggs Type Indicator (MBTI) is still the dominant framework for understanding and assessing personality in software engineering research, we are aware of its poor psychometric characteristics (*Pittenger, 1993*) and that the Five Factor Model (*Digman, 1990*; *McCrae & John, 1992*) is a validated and better choice (*McCrae & Costa, 1989*). The Five Factor model assesses personality along five dimensions, i.e., The Big Five, i.e., extraversion, agreeableness, conscientiousness, neuroticism and openness to experience. The participants' personality was assessed with a validated German version of the Big Five Inventory (*Lang, Lüdtke & Asendorpf, 2001*).

In addition to the mentioned variables, we also kept track of time pressure as a potential confounding factor. After each coding challenge we asked participants on a 5-point agreement scale if they agree that they were under time pressure when solving this coding challenge.

## Analysis procedure

We first report how we assessed solutions to the coding challenges to obtain an overall performance score. Then we describe how we analyzed the data to answer the research question.

### Assessing the coding challenge solutions

The way solutions to coding challenges are evaluated in interviews and programming contests varies, but in the latter there seem to be more objective and absolute judgment criteria than in technical job interviews. It is common for solutions to be evaluated for correctness in programming competitions, and for this purpose there are a couple of test cases for each coding challenge. However, different evaluation schemes have been proposed that differ in the way of assessing solutions which pass *some* test cases (*Kemkes, Vasiga & Cormack, 2006*). For example, the ACM ICPC only differentiates between a correct and an incorrect solution. If any test case fails, then a solution receives no points. The more problems a team solves, the higher it is ranked and in case of a tie, the time needed to solve the problems is the decisive criterion (*Bloomfield & Sotomayor, 2016*). The Google Code Jam works in a similar way, except that each time a participant submits an incorrect solution he or she receives an additional time penalty. In case of a tie, the participant with the lowest penalty time will be ranked first. Other scoring schemes award points for each successful test case. *Kemkes, Vasiga & Cormack (2006)* proposed to award full score for each batch of tests if the solution produces a correct output for *any* test case in that batch. In addition to judging the correctness of a solution, some contests also have a time limit in which a solution has to produce a correct output for a given test case (*Bloomfield & Sotomayor, 2016*). This forces participants of a programming competition not only to find a correct algorithm but also to find an efficient one.

**Table 1  Scoring scheme.** Mapping from a solution's time complexity class to score.

| Challenge 1 | | Challenge 2 | | Challenge 3 | |
|---|---|---|---|---|---|
| Complexity | Score | Complexity | Score | Complexity | Score |
| $\mathcal{O}(n)$ | 1.0 | $\mathcal{O}(n)$ | 1.0 | $\mathcal{O}(n)$ | 2.0 |
| $\mathcal{O}(n \cdot log(n))$ | 0.66 | $\mathcal{O}(n \cdot w)$ | 0.5 | $\mathcal{O}(3^n)$ | 1.0 |
| $\mathcal{O}(n^2)$ | 0.33 | | | | |

To assess a participant's performance we imitated how recruiters evaluate the performance of candidates in technical job interviews, that is in relation to each other and with respect to correctness and time complexity (*McDowell, 2015*). The latter aspect was told the participants before the start of study. Although it would have been possible to determine the best possible time complexity class of an algorithm to the given problems, we could not be sure that this solution could be achieved under the conditions of our study. It was therefore essential to assess the participants' solutions in relation to each other. Additionally, we made use of the *All-or-Nothing scoring* (*Kemkes, Vasiga & Cormack, 2006*) principle known from several programming competitions.

For a participant's solution to a single coding challenge we first run automated test cases on the given source code to see if it produced correct results. If any test case failed, the solution was considered to be incorrect and given zero points. If the solution passed all test cases we analyzed its time complexity. A concrete scoring scheme based on our results is provided in Table 1. Solutions in the best given time complexity class, which, in our scenario, were the solutions with the most efficient algorithm for the problem, were given one point. In case participants came up with more than one correct algorithm in different complexity classes, the solutions were ranked and evaluated on a linear scale between zero and one. This means, for example, that for two correct solutions in different time complexity classes the second best solution receives a score of 0.5, whereas for three correct solutions in different time complexity classes the second best receives a score of 0.66 and the third best solution receives a score of 0.33. As the third coding challenge was expected to be the hardest one, we multiplied the achieved score for the third coding challenge by a factor of 2.0. The overall performance score of a participant was obtained by summing up the scores for each of the three coding challenges. The maximum score achievable was 4.0.

### Data analysis

To answer the research question, we calculated several correlation coefficients between a participant's coding challenge performance score and the quantitative answers to the questionnaire items which operationalized the candidates for predictor variables provided in Fig. 2.

We used correlation coefficient calculations which range from $-1$ to $+1$ to explore which individual characteristics were related to the performance in solving coding challenges. We are aware of an open debate (*Murray, 2013*) on whether Likert scales represent ordinal or continuous intervals. While the debate still does not have clear indications, we opted to consider all our scales to be continuous in nature and Likert items are discrete values on a continuous scale. Therefore, we use Pearson's correlation coefficient where its assumptions

are met, and Spearman's rank correlation coefficient otherwise. We met the latter case for affective states, work experience, experience with coding challenges and the study progress.

We report all calculated correlation coefficients with emphasis on moderate and strong relationships in the following section. The anonymized raw data for this study is available in the Supplements of the present paper.

## RESULTS

We characterize our participants in the 'Methods' section. We refer to them in this section using a post-experiment anonymous identifier in the form of Px, where *x* ranges from 1 to 32. The identifier also represents the ranking of the participants, where 1 implies the highest coding challenge performance score.

We first want to report on the performance of each participant to provide a clear overview of the frequency of time complexity classes for correct solutions and how the overall score for each participant is achieved. Table 1 shows the concrete scoring scheme and Table 2 shows the resulting scores for the participants and illustrates the effect of our scoring scheme. Due to the multiplication factor of 2.0 for the third coding challenge, the weakest solution to the third coding challenge receives the same score as the best solution to the easier coding challenges. However, the ranking order of the participants would not look much different with equal factors of 1.0 for all three coding challenges. P06 would have the same score as P07 to P11. P02 would be ranked after P05 and before P06.

The average participant had 1.72 correct solutions and a score of 1.04 out of a maximum score of 4.0. Eleven participants scored the median and mode value of 0.83. Only one participant achieved the highest possible score and four participants solved none of the challenges correctly and thus had a score of 0.

From Table 2 we see that 28 participants came up with a solution to the first coding challenge but only two participants were able to implement something different from the brute-force algorithm. For the second challenge, of 20 correct solutions, nine run in linear time, which is the best possible for all challenges. Participant P02 as well as P23 to P28 were close to a solution for the second challenge but they did not implement the termination condition for their loop correctly which resulted in failed test cases. The third coding challenge was solved by six participants. Although the number of correct solutions was the smallest of the three coding challenges and the number of complexity classes to which these solutions belong was not higher than for the other coding challenges, with four different algorithms the diversity of correct solutions was the highest.

After each coding challenge, participants had to indicate on a 5-point Likert scale how much they were under time pressure when solving the task they had previously worked on. Accordingly, for the first coding challenge the average time pressure was 2.22 and 1.92 for only those participants who came up with a correct solution. For the second coding challenge the average time pressure was 2.19 and 1.71 for participants with correct solutions. The median value was 2 and the mode was 1 for both coding challenges, which means that participants most often disagreed with the statement that they felt under time pressure. This is different for the third coding challenge for which the average time pressure was 3.84 (median = 4.5, mode = 5) and 2.67 for the six participants with a correct solution.

**Table 2  Performance scores.** Individual performance of the participants. Each row contains the time complexity classes of a participant's correct solution to the corresponding challenge. Incorrect solutions are marked with a dash.

| Participant | Challenge 1 | Challenge 2 | Challenge 3 | Score |
|---|---|---|---|---|
| P01 | $\mathcal{O}(n)$ | $\mathcal{O}(n)$ | $\mathcal{O}(n)$ | 4.0 |
| P02 | $\mathcal{O}(n \cdot \log(n))$ | – | $\mathcal{O}(n)$ | 2.66 |
| P03 | $\mathcal{O}(n^2)$ | $\mathcal{O}(n)$ | $\mathcal{O}(3^n)$ | 2.33 |
| P04 | $\mathcal{O}(n^2)$ | $\mathcal{O}(n)$ | $\mathcal{O}(3^n)$ | 2.33 |
| P05 | $\mathcal{O}(n^2)$ | $\mathcal{O}(n)$ | $\mathcal{O}(3^n)$ | 2.33 |
| P06 | $\mathcal{O}(n^2)$ | $\mathcal{O}(n \cdot w)$ | $\mathcal{O}(3^n)$ | 1.83 |
| P07 | $\mathcal{O}(n^2)$ | $\mathcal{O}(n)$ | – | 1.33 |
| P08 | $\mathcal{O}(n^2)$ | $\mathcal{O}(n)$ | – | 1.33 |
| P09 | $\mathcal{O}(n^2)$ | $\mathcal{O}(n)$ | – | 1.33 |
| P10 | $\mathcal{O}(n^2)$ | $\mathcal{O}(n)$ | – | 1.33 |
| P11 | $\mathcal{O}(n^2)$ | $\mathcal{O}(n)$ | – | 1.33 |
| P12 | $\mathcal{O}(n^2)$ | $\mathcal{O}(n \cdot w)$ | – | 0.83 |
| P13 | $\mathcal{O}(n^2)$ | $\mathcal{O}(n \cdot w)$ | – | 0.83 |
| P14 | $\mathcal{O}(n^2)$ | $\mathcal{O}(n \cdot w)$ | – | 0.83 |
| P15 | $\mathcal{O}(n^2)$ | $\mathcal{O}(n \cdot w)$ | – | 0.83 |
| P16 | $\mathcal{O}(n^2)$ | $\mathcal{O}(n \cdot w)$ | – | 0.83 |
| P17 | $\mathcal{O}(n^2)$ | $\mathcal{O}(n \cdot w)$ | – | 0.83 |
| P18 | $\mathcal{O}(n^2)$ | $\mathcal{O}(n \cdot w)$ | – | 0.83 |
| P19 | $\mathcal{O}(n^2)$ | $\mathcal{O}(n \cdot w)$ | – | 0.83 |
| P20 | $\mathcal{O}(n^2)$ | $\mathcal{O}(n \cdot w)$ | – | 0.83 |
| P21 | $\mathcal{O}(n^2)$ | $\mathcal{O}(n \cdot w)$ | – | 0.83 |
| P22 | $\mathcal{O}(n^2)$ | $\mathcal{O}(n \cdot w)$ | – | 0.83 |
| P23 | $\mathcal{O}(n^2)$ | – | – | 0.33 |
| P24 | $\mathcal{O}(n^2)$ | – | – | 0.33 |
| P25 | $\mathcal{O}(n^2)$ | – | – | 0.33 |
| P26 | $\mathcal{O}(n^2)$ | – | – | 0.33 |
| P27 | $\mathcal{O}(n^2)$ | – | – | 0.33 |
| P28 | $\mathcal{O}(n^2)$ | – | – | 0.33 |
| P29 | – | – | – | 0.0 |
| P30 | – | – | – | 0.0 |
| P31 | – | – | – | 0.0 |
| P32 | – | – | – | 0.0 |

## Happiness

For our participants the average SPANE-P(ositive) value of 22.65 was higher than the average SPANE-N(egative) value of 12.10 and each of the six positive states that contribute to the SPANE-P value were higher on average than their counterparts. The SPANE-B (mean = 11.0, sd = 6.1), 95% CI [8.76–13.24] affect balance score did not differ significantly from the recently established normative scores for the software developer population (*Graziotin et al., 2017*).

**Table 3   Correlation results for happiness.** Summary of the correlation coefficients between the coding challenge performance score and items of the Scale of Positive and Negative Experience (SPANE) for operationalizing happiness ($n = 31$, * $p < .05$, $r_s$ is Spearman's rank correlation coefficient). One participant did not indicate the frequency for all of the affective states. For this reason we dropped his or her record.

| Variable | $r_s$ |
| --- | --- |
| SPANE-B | 0.055 |
| SPANE-P | 0.031 |
| SPANE-N | −0.139 |
| positive | −0.148 |
| negative | 0.166 |
| good | −0.030 |
| bad | −0.237 |
| pleasant | 0.254 |
| unpleasant | 0.013 |
| happy | −0.149 |
| sad | −0.390* |
| joyful | −0.062 |
| afraid | −0.005 |
| contented | −0.052 |
| angry | −0.135 |

Our results show that there is a significant moderate negative relationship between *sad* and the coding challenge performance score, $r_s(29) = −0.390$, $p < .05$. With reference to the discussion in the 'Analysis procedure' section and for the sake of completeness, for this affective state $r(29) = −0.383$. As higher values for the affective states stand for higher frequencies, this negative correlation means that participants who had often been sad in the past four weeks tended to perform worse. The Spearman's rank correlation between *bad* and the performance score is weak negative, $r_s(29) = −0.237$, and additionally there is a weak positive relationship between *pleasant* and the performance score, $r_s(29) = 0.254$. The $p$-values for these two correlations are above the significance level of 0.05. Correlation coefficients for other affective states are given in Table 3.

## Personality

For assessing a participant's personality we used the five factor model which describes a personality by the five traits of extraversion, agreeableness, conscientiousness, neuroticism and openness. Table 4 shows the correlation coefficients between the traits and the coding challenge performance score. What we see from our results is that there is a significant moderate negative relationship between *conscientiousness* and the performance score, $r(30) = −0.352$, $p < .05$. There also is a weak negative relationship between extraversion and the performance score, $r(30) = −0.228$. For the other three personality traits there is no relationship in our data.

## Academic performance

The variables for the academic performance provide the highest values for Pearson's r in our data set. From the results shown in Table 5 we see that there is a strong negative

Table 4 **Correlation results for personality** Summary of the correlation coefficients between the Big Five personality traits and the coding challenge performance score ($n = 32$, * $p < .05$).

| Variable | Pearson's r |
|---|---|
| extraversion | −0.228 |
| agreeableness | −0.058 |
| conscientiousness | −0.352* |
| neuroticism | −0.098 |
| openness | 0.069 |

Table 5 **Correlation results for academic performance.** Summary of the correlation coefficients between the coding challenge performance score and variables operationalizing the academic performance (* $p < .05$, $r_s$ is Spearman's rank correlation coefficient, $n$ is the number of participants for whom a measurement was possible).

| Variable | Pearson's r | $r_s$ | n |
|---|---|---|---|
| Current GPA | −0.448* | | 31 |
| B.Sc. GPA (master students only) | −0.620 | | 10 |
| Grade for *data structures and algorithms* course | −0.557* | | 18 |
| Grade for *algorithms and complexity* course | −0.183 | | 19 |
| Study progress | | 0.179 | 32 |

relationship between the performance scores and the grade point averages master students received in their bachelor's degree, $r(8) = -0.620$. Also, there is a significant strong negative relationship between the performance scores and the grades for the *data structures and algorithms* course, $r(16) = -0.557$, $p < .05$ and a significant moderate negative relationship between the performance scores and the current grade point average, $r(29) = -0.448$, $p < .05$. As in Germany lower grades are better than higher ones, these negative relationships mean that participants with better grades were also the better coding challenge solvers.

The grade point average for the *data structures and algorithms* course was 1.66 (sd = 0.79) in our sample and therefore much better than the grade point average for the *algorithms and complexity* course which was 3.11 (sd = 0.83). We only see a weak negative correlation for the latter course with the coding challenge performance score, $r(17) = -0.183$.

Study progress was represented as a three-valued factor: students at the beginning of their bachelor's program (14 participants), students that are at least in the fourth semester of the bachelor's program (7 participants) and master students (11 participants). The participants in the first group had an average score of 0.66 (median = 0.83). Those in the second group had a score of 1.89 (median = 1.83). The third group had an average score of 1.07 (median = 0.83). The highest score of 4.0 was achieved by a master student, the maximum score in the group with the advanced bachelor students was 2.66 and the maximum score for the students at the beginning of their bachelor's program was 1.83.

We found a weak positive relationship between the study progress and the performance scores, $r_s(30) = 0.179$.

**Table 6 Correlation results for experience.** Summary of the correlation coefficients between experience and the coding challenge performance score ($n = 32$, * $p < .05$, $r_s$ is Spearman's rank correlation coefficient).

| Variable | Pearson's r | $r_s$ |
|---|---|---|
| Coding challenge experience | | 0.227 |
| Programming experience | 0.420* | |
| Java experience | 0.184 | |
| Work experience | | 0.196 |

## Experience

In the last part of the questionnaires, we asked the participants experience-related questions. Programming experience, experience with Java, and experience with working in a company with focus on software development were measured in years. Experience with coding challenges in the past year was indicated by the participants on a 6-point frequency scale.

From Table 6 we see a significant moderate positive relationship between the coding challenge performance score and the programming experience of a participant, $r(30) = 0.420$, $p < .05$. On average, participants had 5.02 years of programming experience (sd = 2.47). For the work experience, coding challenge experience and the experience with Java we only observed weak positive relationships with the performance score. 21 participants answered that they never had experience with coding challenges in the last year, five participants did coding challenges once or twice per semester, five participants did them once or twice per month and one had experience with coding challenges once per week. When we asked the participants afterwards what their concrete experience with coding challenges was, they mainly told us about exercises they had to do for the *algorithms and complexity* class and that these exercises were pretty similar to the coding challenges we used in our study. The one participant who indicated to solve coding challenges once a week told us that he or she solves them for fun on the Internet. This participant had the second best coding challenge performance score of 2.66, while the participant with the best performance score of 4.0 had not done any coding challenges in the past year.

We finally asked participants for their open-source profiles and their Stack Overflow profile to explore the contributions to the respective communities and see how they correlate with the scores in the coding challenge performance. Of the 32 participants only four participants had a Stack Overflow profile, three of whom have contributed at least one question or answer to the network. The coding challenge performance scores for these three participants were above average, but their contributions were made mainly to fields unrelated to Java, algorithms or programming puzzles. More participants provided us with a URL to their GitHub or GitLab profiles and eight of them contributed at least one public pull request, but we did not observe a relationship between the number of pull requests and the performance score. The eight participants with public open-source contributions had an average performance score of 1.16, which is slightly higher than the average performance score of 0.98 for participants without public open-source contributions. The majority of projects they contributed to were programmed in Java or JavaScript.

# DISCUSSION

## Findings

We found some significant correlations between individual characteristics and the coding challenge performance.

First, we would like to state that we are aware that reporting *p*-values in exploratory studies is potentially problematic (*Rubin, 2017*; *Neuroskeptic, 2015*) because of the open debate of what these *p*-values really represent (e.g., a null hypothesis significance testing of an absence of relevant factors in the first place). The discussion is so recent that we opted for the conservative choice to use *p*-values and their classic value for significance ($p < .05$) as a threshold to include (or exclude) variables in our theory. Our theory provides relationship links—which are of correlation type and not causality—and indications for the polarity of these relationships. The theory, however, does not include numerical assessments of the strength of these relationships. They are outside the scope of an exploratory study towards a Type III theory.

### Happiness

In the 'Conceptual model' section we justify happiness as a candidate predictor variable by findings on the positive impact of happiness and affect on job performance and analytic problem-solving. For example, one finding by *Graziotin, Wang & Abrahamsson (2014)* was that happy software developers perform significantly better in analytic problem-solving. In our study, we could not find a positive correlation between SPANE-B and the coding challenge performance, $r_s(29) = 0.055$. We only observed a weak positive relationship between the positive affective state *pleasant* and the performance, $r_s(29) = 0.254$. However, we found that *sad* software engineers performed significantly worse, $rs(29) = -0.390$, $p < .05$.

We believe that there could be a cause–effect relationship between sadness and the coding challenge performance because of the participants who felt sometimes or often sad in the past four weeks (approximately 28%), nobody came up with a correct solution to coding challenge 3. Furthermore, for the first two challenges, no one came up with an algorithm different from brute-force solutions. Consequently, none of the sad participants had a score higher than 0.83.

One possible explanation for the misalignment between our study results and those of the *Graziotin, Wang & Abrahamsson (2014)* study results could be that coding challenges constitute an atypical programming task and, therefore, the performance in coding challenges does not necessarily coincide with software development performance. We offer this speculation for future studies to explore.

### Personality

The personality trait *conscientiousness* showed a significant moderate negative relationship with the coding challenge performance score, $r(30) = -0.352$, $p < .05$. This means the higher the score for *conscientiousness*, the lower the coding challenge performance score. Conscientiousness describes the extent to which a person is a reliable worker, perseveres

until the task is finished, does a thorough job and does things efficiently. A conscientious person is not careless, does not tend to be disorganized or lazy, and is not easily distracted.

To understand the relationship between the performance and the personality trait we inspected the answers to the statements of the Big Five inventory of the six participants with the highest performance score. Interestingly, they tended to be reliable workers, which increased their score for *conscientiousness*, and they did not tend to be easily distracted, which would have lowered their score for the personality trait. Their average score for *conscientiousness* was 4.17, which is only slightly below the average value of 6.69 for the rest of the sample. Beside Pearson's $r$, we also always considered Spearman's rho to avoid wrong assumptions due to the possibly strong influence of outliers and sequences which are not entirely homoscedastic. Still there was a negative monotonic relationship, $r_s(30) = -0.294$. Although we cannot explain the relationship, based on our findings, we would like to state the hypothesis that conscientious persons perform worse in coding challenges and let future research examine this relationship in more detail.

As existing literature found particular personality types to positively correlate with programming task performance (see *Cruz, Da Silva & Capretz (2015)* for their systematic mapping study) and even that conscientious programmers perform better in programming tasks (*Karimi et al., 2016*) we would like to repeat our speculation that coding challenges may differ from ordinary programming tasks and invite future studies to investigate the differences.

For the other personality traits of the five factor model, i.e., *extraversion*, *agreeableness*, *neuroticism* and *openness*, we did not observe any significant correlation with the coding challenge performance.

### Academic performance

We found moderate to strong linear correlations between two GPA-related variables and the performance score. The significant moderate negative relationship between the current GPA and the performance score, $r(29) = -0.448$, $p < .05$, shows that students with better grades performed better in the coding challenges. Additionally, the Pearson's correlation coefficient between the B.Sc. GPA and the performance score was strongly negative, $r(8) = -0.620$, but due to the small number of master students, the relationship could have occurred by chance ($p = .056$).

Many of the bachelor students at the end of the second semester mentioned that their current GPA consisted only of one or two grades. As this group of students made up about 44% of our sample, this should be taken into account. However, because we observed more negative relationships for grade-related variables, it can be reasonably assumed that we would have observed a negative relationship also if students had taken more than one or two exams.

There was only a weak positive correlation between study progress and the performance score, $r_s(30) = 0.179$, and we observed that the group of higher bachelor semesters performed best. In discussions after the study, participants told us that in the *algorithms and complexity* course, students nowadays have to solve tasks similar to the coding challenges we used in our study. As this course is usually taken in a higher bachelor semester, this

could have been the reason why they performed better on average than both the younger bachelor students and even the master students since their knowledge was still fresh.

Furthermore, as students at the beginning of their bachelor's program performed worst and master students did not perform much worse than students in a higher bachelor semester, we assume that there is a baseline of knowledge one has to be aware of when solving coding challenges but further academic progress does not necessarily make a coding challenge solver better.

Taking all this, and especially the previous paragraph, into consideration, we cannot fully assume that receiving better grades leads to better coding challenge performance. But we can speculate that there is a confounding variable that predicts how well a student performs in his or her exams and how well he or she performs in solving coding challenges.

Further, our results show a significant strong negative relationship between the grade for the *data structure and algorithms* course and the performance score, $r(16) = -0.557$, $p < .05$. Again, this negative correlation means that students with better grades were the better coding challenge solvers. For the weak negative relationship between the grade in the *algorithms and complexity* course and the performance score, $r(17) = -0.183$, we would like to note that some of the participants were examined by a different examiner and they told us that therefore the exam became easier.

A good understanding of data structures and algorithms is fundamental for finding an efficient algorithm to a given coding challenge. Therefore, we assume that a good preparation for the *data structure and algorithms* exam not only leads to a good grade but also improves the coding challenge performance. Taking this finding one step further, it provides at least an indication of how the targeted preparation for solving coding challenges could have an impact on the coding challenge performance.

### Experience

For the experience-related variables we observed a weak positive relationship between the coding challenge experience and the performance, $r_s(30) = 0.227$, between the Java experience and the performance, $r(30) = 0.184$, as well as between the work experience and the performance, $r_s(30) = 0.196$. The weak correlation coefficient for the Java experience could be explained by our selection of coding challenges which do not require specific knowledge of the programming language. The positive relationship between the experience with coding challenges and the performance in solving such is in line with what *Revilla, Manzoor & Liu (2008)* found. They conclude, from statistics of an online judging system, that solving more coding challenges increases the individual acceptance rate and decreases the rate of wrong answers as well as of compilation errors, while the rate of submitted solutions that exceed a given time limit doesn't change. For problems with a low acceptance rate the wrong answer rate almost remained the same, independently of the number of problems a user solved. *Bloomfield & Sotomayor (2016)* claim that the biggest success factor for programming competitions are training activities like working through problems and running team sessions in which problems are discussed. Unfortunately, they do not provide evidence other than by reporting their own experience. Although in our sample the best coding challenge solver did not have any experience with coding challenges in the past

year and the overall correlation between coding challenge experience and coding challenge performance is weak, we recommend future research on the effect of targeted preparation for solving coding challenges. In our sample very few people have had experience with coding challenges other than in assignments of university courses.

More importantly, we observed a significant moderate linear relationship between the years of programming experience and the coding challenge performance, $r(30) = 0.420$, $p < .05$. We justified the inclusion of our variables with existing literature, so we expected to find a relationship between experience and coding challenge performance. However, the particular correlation between programming experience in years and a software engineer's performance in solving a programming task was not consistently observed to be positive in the past. For example, *DeMarco & Lister (2013)* did not find a correlation between the years of programming experience and the performance in terms of time to complete a programming task, at least not for participants with more than six months' experience. Those participants with less than six months' experience performed worse than the rest. The contradictory results could be due to the difference in the programming tasks and the different definitions of performance. Working as fast as possible on an ordinary programming task arguably requires different skills than finding an efficient algorithm to a coding challenge.

According to our results we believe that an increase in programming experience leads to a better coding challenge performance. This might only hold true until a threshold is reached but it seems to be greater than the six months observed by *DeMarco & Lister (2013)*.

## Theory for predicting the performance in solving coding challenges

The proposed theory for predicting the performance in solving coding challenges is provided in Fig. 3. We theorize that individual characteristics, such as happiness, personality, academic performance, and experience are predictive for the performance in solving coding challenges. Our theory is meant to be what *Gregor (2006)* calls a Type III theory. We already gave a brief overview and definition of this theory type in the 'Background' section. We used the work of *Gregor (2006)* as a framework to build and represent our theory. We will now use the framework as a benchmark to discuss how each of the structural components of theory is present in our work.

*Means of representation* is defined as the physical representation of the theory (*Gregor, 2006*: 620). Our theory is represented by words and a diagram (Fig. 3).

*Constructs* "refer to the phenomena of interest in the theory" and "all of the primary constructs in the theory should be well defined" (*Gregor, 2006*: 620). Primary constructs of our theory are coding challenges, the coding challenge performance, and the four constructs we refer to as individual characteristics, namely happiness, personality, academic performance and experience. We defined the term coding challenge in the 'Background' section as an algorithm and coding problem used to assess a programmer's problem-solving skills. The performance in solving coding challenges is obtained by aggregating the individual scores for solutions to each one in a set of coding challenges. We describe the scoring algorithm for a single coding challenge and the aggregation algorithm in the 'Analysis procedure' section. Algorithm complexity is a highly relevant concept for

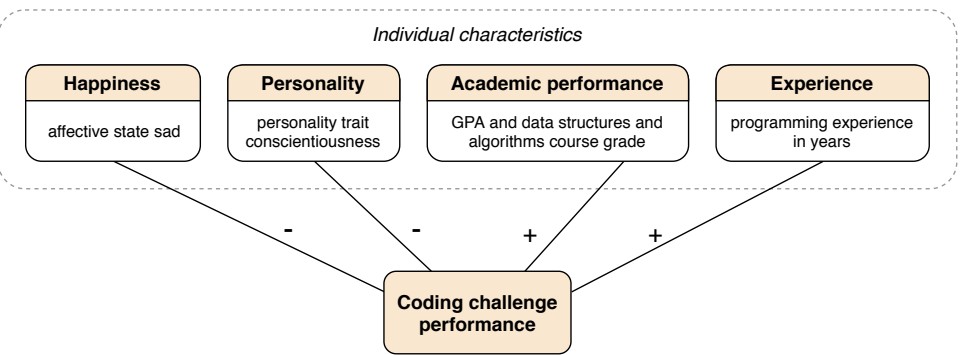

**Figure 3** **Summary of the obtained theory for predicting the performance in solving coding challenges.** Connecting lines depict a significant positive or negative correlation between the coding challenge performance and operationalizing variables for each of the four constructs of individual characteristics.

obtaining the coding challenge performance score, but we see this construct covered by the definition of the coding challenge performance. Happiness, personality, academic performance and experience have all been defined in the 'Conceptual model' section.

*Statements of relationship* "show relationships among the constructs" (*Gregor, 2006*: 620). Correlations are the degree of relationship between variables of individual characteristics and the coding challenge performance. The affective state of sadness is negatively correlated with the performance in solving coding challenges. The same applies for the personality trait of conscientiousness. GPA and the grade in the *data structures and algorithms* course are positively correlated with the performance in solving coding challenges. The same applies for programming experience in years.

*Testable propositions* as theory component describes the necessity that statements of relationships can be tested empirically (*Gregor, 2006*: 620). The statements of relationships in our theory could be tested. We obtained them empirically using proven statistical methods. Furthermore, the paper presents testable propositions of the theory to be further tested by future work.

*Scope* "is specified by the degree of generality of the statements of relationships (signified by modal qualifiers such as 'some,' 'many,' 'all,' and 'never') and statements of boundaries showing the limits of generalizations" (*Gregor, 2006*: 620). The previously defined statements of relationships include no modal quantifiers. We do not believe limitations of our study (see the related section) to have an influence on the scope in which our statements of relationships hold true. It might be the case that in a different context, further variables such as the stress level or the sympathy for the candidate might have an influence on the rated performance. These additional variables could be significant but are believed to not invalidate our statements of relationships. The theory might therefore be refined and extended by further individual characteristics and by contextual variables in the future.

*Causal explanations* and *Prescriptive statements* are not present due to the nature of Type III theories.

[4]We are grateful to two anonymous reviewers for going above and beyond in suggesting challenging and interesting limitations of the study

## Limitations

The study design, as described in the 'Methods' section, has some limitations which have to be considered when interpreting the results and defining the scope of our proposed theory. These limitations are about the sample, the adherence to real-world settings, the design of the challenges, and the elements of the theory itself.[4]

The sample was limited in size and consisted solely of students, all of which were BSc and MSc students from a German university. This limits the way we can generalize our theory as the sample might poorly represent students worldwide as well as job seekers. We believe, however, that our sample, while limited in terms of nationality and place of study, represented the desired population fairly, that is potential hires of successful IT companies. We also ensured that the sample did cover academic status well as we invited students whose academic progress ranges from the beginning of the bachelor's program to the end of the master's program.

We could not ensure that the settings of our study would fully reflect the settings of a job interview, especially in terms of stress and anxiety and participants' preparation. In a real-world setting, a job position is at stake and the interviewee is not anonymous and faces the interviewer. In case the candidate has to write source code on a whiteboard, stress and cognitive load are additionally reinforced (*Behroozi et al., 2018*). Instead, in our study we tried to create a relaxed atmosphere and that is what an interviewer should do. Future work can examine the performance of candidates in real technical interviews and assess the individual characteristics of the candidates. Similarly, our participants did not prepare for solving coding challenges in the past. In real-world settings, a candidate will ideally prepare for the interview, including the coding challenges part. We coped with this issue by operationalizing the variable *experience with coding challenges*. Not much research has been done yet to examine the effect of targeted preparation for coding challenges so we do not know about its impact and how well our results can be applied to a group of software engineers who all recently prepared for programming interviews or programming competitions. The issue is also less threatening for us as solving the challenges we chose does not require knowledge of particular programming language features or the like that an unprepared software engineer would not have. Future research can repeat our study with prepared participants and see if and how the results differ.

Our choice of tasks and their scoring reflects real-world settings as closely as possible, coming all from the related literature, yet it carries several assumptions that we feel should be clarified here. First, we did not assign partial scores for correctness nor did we take into account the quality of the code being produced. As mentioned in section 'Assessing the coding challenge solutions' and further discussed by *Kemkes, Vasiga & Cormack (2006)*, different strategies for assessing computing competition tasks have existed for some time. We opted for the *All-or-Nothing* principle because we have hardly touched any edge or special cases with our software tests, and we have already excluded some special cases by limiting the input parameters in the task description. Consequently, solutions were only marked as incorrect if they really did not correctly implement the basic functionality required. We opted to lose information as the price to pay to ensure objectivity in the assessment. We will elaborate more on this after introducing another potential issue, that

is that we did not consider the quality of the code being produced. We recognize that code quality is relevant, to a certain extent, when programming occurs in an interview. *McDowell (2015)* lists the following properties for *good code* that employers want to see: correct, efficient, simple, readable and maintainable. While we looked at the first two properties, correctness and efficiency, we did not consider simplicity, readability, maintainability or other code quality characteristics when we evaluated the solutions. We chose problems to which possible solutions usually should be around ten to twenty lines long, not expecting much variability in the produced code. We also encouraged participants to focus on correctness and efficiency so that our evaluation process was as objective as possible.

On our two reasons to opt for the previously mentioned tradeoffs for gaining in objectivity. We performed a sensitivity analysis to establish what would happen if we gave up some objectivity and assigned points for recognizably correct approaches with incorrect implementations. We inspected solutions for which at least one test case failed. For the first challenge, all four incorrect solutions were completely wrong so that we would not have awarded partial points for any of them. For the second challenge, we received 11 incorrect solutions. P02, P23, and P28 were close to a correct solution and we could have awarded partial points for a correct approach. If we had done so, they would have received a score less than 0.5 (less than the score for a correct solution) in the worst time complexity class for this challenge, and this would not change the final ranking of the participants. This is because P23 to P28 would then still have a lesser score than P22. P02 would still be placed second. For the third challenge we find it very difficult to evaluate most of the 26 incorrect approaches scoring 0 points for their partial correctness. They all fail in most test cases and approach the problem in very different ways, but we cannot say how much extra effort would have been needed to arrive at a working solution. For almost half of the incorrect solutions, we concluded that they would not deserve partial points, either because there is no implementation at all or because participants found the solutions to a few values of the input parameter *n* manually and return them in an if-else-cascade. The other half of the solutions are difficult to compare with each other or with one of the sample solutions so that we cannot objectively give partial points to them at all.

The above reasoning on assigning points only to correct solutions and not to judge code quality might introduce a further potential threat to the design of our study, that is that one might conclude that on average, a candidate should strive to answer all questions with suboptimal times; otherwise, attempting to find the ideal time complexity (and failing) could result in performing worse. In parts we agree to this assumption but we do not see it to be severe in the challenges we opted to use. For the first task, the suboptimal solution was arguably very obvious and after the implementation there might not have been enough time to think about the optimal algorithm. Our guess is that most participants would not have come up with the ideal solution even if they had been told what the minimum time complexity is and that they should try to implement such a solution. Our guess is based on informal discussions with the participants that we had after the study and in which we discussed the optimal solution. For the second task, we see no strong difference in effort between the suboptimal solution and the ideal solution, as we believe that only realizing

the optimal solution would drive the participant to its implementation. For the third task, most of the participants already had issues in finding any correct solution at all. Some of the suboptimal solutions we received for this task were also much more complex to implement than the solutions presented in the manuscript. When introducing the participants to the study, we explicitly told them that there was no advantage in submitting a solution before time was up and that they could improve their solution with any remaining time. So even if they had started with a _brute-force_ solution, they would have had a chance to improve their solution. We also recorded the time taken to complete a task. From this data and from their responses to the perceived time pressure, we see that many of the participants had completed at least the first two tasks before the available time had elapsed.

We would like to make a few observations on the actual results that we obtained. As one would expect, our 32 participants did not perform equally good in our three tasks. The ability to solve the challenges decreased as difficulty increased (see Table 2). The third and most difficult challenge was solved by only six participants, only two excelled with it and, overall, only one participant obtained the highest possible score. The situation of having one top performer and other five fairly good performers might appear to be a limitation, as it might appear that tasks were too difficult or even that coding challenges are not a suitable tool for job interviews as they are likely to yield high performers. This is by design, and we did not expect many participants to excel overall. Challenge 3, in particular, was intended to separate the really good participants from the average ones. The design of the tasks follows guidelines for coding challenges from the literature ('Coding challenges' section), so the coding challenges are tasks that companies adopt for pre-interviews and interviews. Our environment and task successions, while artificial, are supposed to adhere to the real-world situation. The only deviation was that sometimes companies adopt even more tasks than we did, and we could not demand more time from our participants to add more tasks. While it would have been more scientifically interesting to end up with some more top performers, this did not happen, and only repetitions of this study would be able to tell us if that was by chance.

Finally, the elements of our conceptual model are just one possibility for forming an initial theory on individual characteristics of coding challenge performance. Many other factors were not included in the model, including but not limited to cognitive processing abilities, further tests for knowledge of the involved domains, and salient characteristics such as sleeping time. To recap, we opted to derive factors from psychology, education, and software engineering literature that have been suggested to correlate with (or cause) performance that is similar to the one we look for. The factors we included are also easily verifiable at interview stage by companies. An alternative to this approach would have been to conduct an exploratory qualitative study to derive a richer set of factors from the experiences and perceptions of participants. However, such a type of study, while interesting and welcome for future work, would have prevented us from providing an initial evaluation of which of those factors are likely to have an influence on the dependent variable. We see that coding challenges are a central and pressing topic in software companies, startups and corporations alike, and we believe that a first quantitative, yet deeper, quantitative exploration brings interesting and practical results than a qualitative, broader yet shallower

exploration. Ultimately, we believe that the robustness of our methodology which closely follows an established framework for theory building, classification, and representation, comprises strong building blocks for future validation studies as well as studies to identify further factors and enrich our theory.

## Implications

The theoretical implications of our study lie in the theory itself. We constructed the theory systematically by inspecting the literature to construct a conceptual model and then performed a first validation of the model in an exploratory study. The theory is thus grounded in empirical data. We request future studies to validate the theory under different settings and samples, as well as extend it with further constructs that we could not include in the present study. Our theoretical contributions provide basic building blocks in the body of knowledge of software engineering research.

The practical implications of our theory are limited, yet the results of the study are interesting to software companies and practitioners, should they be generalized.

Each company that assesses a candidate's performance with coding challenges should be aware of the negative correlation between the personality trait of conscientiousness and the candidate's performance. There is a need for personality diversity in software engineering, among other things, because "there is no single personality type that fits the wide spectrum of tasks that encompass the engineering of software" ([11], p. 10). Some studies showed a relation between personality diversity and team effectiveness, others showed the contrary (*Cruz, Da Silva & Capretz, 2015*). Conscientiousness is one of a few personality traits which are believed to positively correlate with a software development team's effectiveness (*Yilmaz et al., 2017*) and individual satisfaction (*Acuña, Gómez & Juristo, 2009*). As a consequence, the decision for or against the use of coding challenges for the evaluation of a job candidate's performance should take the personality diversity of the other team members into account.

In case coding challenges have to be solved—whether in technical interviews or in programming competitions—interviewers and organizers should always be interested in the well-being of the coding challenge solvers. This can be seen as a general recommendation regardless of our results which indicate a negative correlation between the affective state "sad" and the coding challenge performance. Google, for example, measured each candidate's experience with their interview process and found this to be correlated with the proportion of candidates who accept the job offer as well as the percentage of rejected candidates who would still recommend the company to a friend (*Google Inc., 2018*). However, even when taking care of the overall experience with the interview process, the comparison of two job candidates based on their performance in solving coding challenges could be biased due to differences in their happiness. This might not easily be accessible or directly improvable. A corrective action a company could apply is to give a second chance to candidates for whom the interviewer felt that they would perform better at some other point in time. This would minimize the chance of rejecting actually well fitting candidates who performed badly due to disadvantageous affective states.

If a company is in the fortunate situation of having more applicants for a position than the company can assess via on-site interviews using coding challenges, then academic

performance, i.e., GPA and the data structures and algorithms course grade, are good criteria for preselecting candidates based on their resume. GPA is already frequently used as biographical data item for screening applicants as for recruiters, the GPA represents personal motivation as well as language and mathematical capacities (*Brown & Campion, 1994*). A better GPA results in more first choices and even the decision to list the GPA as biographical information on a resume will do so (*Thoms et al., 1999*). Our results imply that there is nothing wrong with that with respect to the intention of selecting appropriate candidates for more expensive on-site coding challenge interviews based on their academic performance. From a candidate's point of view, acquiring knowledge on data structures and algorithms is beneficial for both better course gradings and a better coding challenge performance.

Our results further imply that programming experience is also a good criterion for preselecting candidates. Although programming experience in years is something which is usually not directly stated on a resume, experience can be derived from several other resume items. For example, *McDowell (2015)* advises to build and contribute to projects because having them on the resume "is often the best way to present yourself as more experienced. This is especially true for college students or recent grads" (p. 28).

We do not feel that our results indicate to which extent coding challenges are an effective tool for interviewing or even pre-screening job applicants. As we explain in the 'Limitations' section, we ended up with a situation of less than 20% of the participants performing well overall and only less than 3% of them (one participant) being a top performer. Coding challenges are capable of filtering out many candidates. While we welcome future studies to investigate whether coding challenges are an effective tool for candidate selection, we urge companies to bear in mind that an absolute score in a coding challenge should not be the only decisive factor in finding good candidates.

## CONCLUSION

The ability to be a successful coding challenge solver is essential in many technical interviews. Yet little research has been conducted on predictor variables for a candidate's performance, which results in a failure to understand why some people perform better than others in solving coding challenges and ultimately biases hiring decisions.

This paper started to fill this research gap by investigating individual characteristics of successful coding challenge solvers. We reported on an exploratory study towards a theory for predicting the coding challenge performance with four constructs, namely happiness, personality, academic performance and experience. It became evident that the affective state *sad* as well as the personality trait of *conscientiousness* negatively correlate with the coding challenge performance. GPA, the *data structures and algorithms* course grade, as well as the programming experience in years positively correlated with the performance.

Recruiters and interviewers can take these findings into account when they screen resumes and decide for coding challenges as a means for measuring a candidate's skills. Being aware of possible predictor variables can reduce hiring costs and bias in hiring decisions by taking suitable measures as we discussed earlier in this work.

As we observed a difference between some of our results and the results of previous studies on the relationship between individual characteristics and software development performance, we speculate that coding challenges constitute an atypical programming task. We offer our observations for future studies to better understand them.

Moreover, we offer the theory to be tested in future work. Due to the exploratory nature of our study, the observed relationships can be used to establish hypotheses which future work can test. The theory can then be extended, for example, by further individual characteristics and by contextual factors. Taking some of the limitations into consideration, future studies can be designed to be more similar to technical interviews and could conduct such a study with prepared candidates to see how the results differ. Finally, obtaining causal explanations for the relationships might enable the theory to be classified as a theory for explanation and prediction. A better understanding of the underlying causes allows sound recommendations for actions to be made which practitioners can benefit from in the future.

## ACKNOWLEDGEMENTS

We gratefully acknowledge the students who took the time to participate in our study and Kornelia Kuhle for proofreading the work.

### Funding

Daniel Graziotin was supported by the Alexander von Humboldt (AvH) Foundation. The funders had no role in study design, data collection and analysis, decision to publish, or preparation of the manuscript.

### Grant Disclosures

The following grant information was disclosed by the authors:
Alexander von Humboldt (AvH) Foundation.

### Competing Interests

The authors declare there are no competing interests.

### Author Contributions

- Marvin Wyrich conceived and designed the experiments, performed the experiments, analyzed the data, contributed reagents/materials/analysis tools, prepared figures and/or tables, authored or reviewed drafts of the paper, approved the final draft.
- Daniel Graziotin conceived and designed the experiments, analyzed the data, contributed reagents/materials/analysis tools, prepared figures and/or tables, authored or reviewed drafts of the paper, approved the final draft.
- Stefan Wagner conceived and designed the experiments, contributed reagents/materials/analysis tools, prepared figures and/or tables, authored or reviewed drafts of the paper, approved the final draft.

## Data Availability

The raw, anonymized data and the data analysis of the study are available in the Supplemental File.

## Supplemental Information

Supplemental information for this article can be found online at http://dx.doi.org/10.7717/peerj-cs.173#supplemental-information.

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
