# Peer review of "A theory on individual characteristics of successful coding challenge solvers"

_PeerJ Computer Science, doi:10.7717/peerj-cs.173_

## Round 0.1 · original submission · Major Revisions

Dear authors,

thank you for your valuable submission to PeerJ CS. After careful review, both reviewers agree that the study subject is interesting and important. However, particularly Reviewer 2 has raised some interesting questions.

My impression is that fully addressing all of these questions may be infeasible without re-executing the entire study - which in my opinion is not necessary as the current incarnation already provides many insights. However, I hope for a more thorough discussion of experimental limitations and factors that are not controlled for in the study in the revised version of the manuscript.

Reviewer 1 ·

Basic reporting

The paper offers a clear outline of the proposed research and its underpinnings; although I missed an initial series of hypotheses to be tested, which is typical when correlation analysis is used as a theoretical building framework, the work is well elaborated and offers interesting insights. On the one hand the paper is well connected to state of the art literature but on the other hand some related work in software engineering skills, education, and understanding is still missing, e.g., [1] and [2] off the top of my head. Furthermore, from yet another perspective, several early attempts were made at offering a "goodness index" for software developers (e.g., [3]) - these should at least be mentioned in the scope of the paper as related work.


[1] Teles, V. & de Oliveira, C. (2003). Reviewing the Curriculum of Software Engineering Undergraduate Courses to Incorporate Communication and Interpersonal Skills Teaching. Software Engineering Education and Training. Proceedings. 16th Conference on (p./pp. 158--165), Los Alamitos, CA: IEEE Computer Society.

[2] Matturro, G. (2013). Soft skills in software engineering: A study of its demand by software companies in Uruguay.. CHASE@ICSE (p./pp. 133-136), : IEEE Computer Society. ISBN: 978-1-4673-6290-0

[3] Borreguero, F., Nitto, E. D., Stebliuk, D., Tamburri, D. A. & Zheng, C. (2015). Fathoming Software Evangelists with the D-Index.. In A. Begel, R. Prikladnicki, Y. Dittrich, C. R. B. de Souza, A. Sarma & S. Athavale (eds.), CHASE@ICSE (p./pp. 85-88), : IEEE Computer Society. ISBN: 978-1-4673-7031-8

Experimental design

The design of the experimentation appears sound although, as the authors also notice, the experimentation can and should only be considered preliminary, since the sample population is missing some obvious control factors (e.g., age, background, etc.) - that being said, it would be sufficient to elaborate on these aspects in the threats to validity section (perhaps discussing which areas of validity are affected by which sampling assumptions) and in the future work section.

Validity of the findings

the findings were very interesting although visualized in a preliminary and basic way; more advanced visualization plots may have helped further understanding of the work and its implications, perhaps even helping with the discussion.

Additional comments

The title is a bit convoluted - I would choose a new one based on the actual goals that the authors are exploring, e.g., "Developer Characteristics vs. Coding Challenges: an Empirical Study"; in essence, anything more indicative of the actual content of the work would help.

the paper has several imprecisions in English grammar, spelling and some haphazard ways of presenting table entries (e.g., table 3 and following, please use "p-value" instead of "p", explain "n") - these issues should be addressed to maximise paper quality.

Reviewer 2 ·

Basic reporting

This article presents a study of 32 students who solve 3 coding challenges. Several factors, such as personality traits, and exam grades are used to construct a theory for predicting coding challenge success.

Overall, I admire the careful construction of study, detailed discussion of literature, and clarity in writing and presentation of results. However, there are a few issues that stand out in experimental design and interpretation of the results. Some of these issues could be addressed with careful discussion and improved framing of the results. However, some of these issues also lead me to question the validity of the results and purpose of the study itself.

In conclusion: I worry that a really nice and careful study was performed without really moving forward our understanding on this problem and I'm not sure how to fix this without considerable reframing or additional investigations.

Experimental design

### Single High Performer

Out of 32 participants, only 1 participant succeeded in answering all three questions with the ideal time complexity. The remaining participants are split between answer all or most questions with suboptimal time complexity, or not succeeding at all. There are several interesting consequences/observations:

* Perhaps the questions were simply too difficult for the given population. Students that typically enroll in code challenges may be atypical, or practice/train for competitions. What would have happened if there were more questions like "countPairs".

* Several of the problems involve considerable problem translation (_build a mansion along a road, other side of road are houses, in which a certain number of people live_). These can be quite strange and abstract, leading to misinterpretation and significant time trying to understand what the problem is asking. Interesting enough, the proposed ideal solution is not very complex (a simple for loop), suggesting these are problem statements may be intended to serve as a distractor.

* What was the main reason to count the solution to be completely incorrect by failing on a test case? How would counting partial solutions change the results or conclusions?

* Given the skewed distribution of results (with only one fully correct and ideal set of solutions), it is difficult to conclude anything about what makes a "successful" coding challenge solver. Indeed, if that was truly the goal, then it would have been better to study students who are engaged in the competitions themselves and not a random population.

* From a "game theory" perspective, one might conclude that on average, a candidate should strive to answer all questions with suboptimal times; otherwise, attempting to find the ideal time complexity (and failing) could result in performing worse.

* We could also conclude the "coding challenges", especially the nature used in the study, are unsuitable for interviews since they are unlikely to yield high performers. I would have to interview 32 candidates in order to potentially find one high performer.

Validity of the findings

### Theory

Four factors were investigated in constructing a theory: Happiness, Personality, Academic Performance, and Experience.

The coding challenges did not exhibit any need to used data structures (trees, graphs, hashtables, hashmaps, linked lists etc.). It is not clear what content has prepared the students for these coding challenges and can simply related to some other latent variable. One possible alternative conclusion can be that success in Algorithms and complexity course is more representative than success in Data structures and algorithms course --- because students have more exposure to the concept of time complexity and also receive training on code challenges.

Emotional state was measured.
> We believe that there could be a cause-effect relation between sadness and the coding challenge performance, because from the participants who felt sometimes or often sad in the past four weeks, nobody came up with a correct solution to coding challenge 3.

Are there sad because they are not doing well in school? Or because they were incidentally not having a good day, recently experienced a stressful event, or maybe they were just tired [Fucci 2018]. Is this a residual effect or ephemeral effect?

In short, it is not clear how well connected or justified any of these characteristics are to the 3 coding challenges and larger research question being studied. Why are other factors such as sleep, word problem comprehension, typing speed, Java experience, interview experience, also not investigated?

It is not clear we have the appropriate elements for building a theory.

[Fucci 2018] Need for Sleep: the Impact of a Night of Sleep Deprivation on Novice Developers’ Performance

Additional comments

### Clarifications/Minor

* Explain conscientiousness. What is this. What is the significance and implications of this personality trait. What is the natural frequency of this trait in human populations.

* Do they have the same examiner in Data structures course?

Line: 202 "ideal employable age" sounds like a lawsuit in waiting. Rephrase.
Line 250: How did you made sure that the pilot candidate is a good representative of their whole candidate population?
Line 252: How did you verify candidates were expressing time pressure?
Line 366: Redundant word “found”
Line 379: What are those strong reasons?
Line 611: Did they want to say “junior” instead of “senior”?

---

## Round 0.2 · accepted · Accept

I agree with the reviewer that the manuscript is now in shape to be published. Congratulations!

Reviewer 2 ·

Basic reporting

The writing is quite clear and precise. Sufficient literature references are provided. Raw data is shared for replicating the calculations. The paper is self-contained.

Experimental design

The experiments are well designed and described.

Validity of the findings

The authors are generally careful in delineating conclusions based on evidence and speculation.

Additional comments

While there were limited actions the authors could take to address some of the questions raised in the initial review, the revised version of this article has gone through great lengths to improve the limitations, discussion, and add clarity to the paper, nearly 2 more pages of new content in total. In short, I believe this currently paper stands well on its own, even with its limitations. My hope is that these questions will help guide future work.